# Neurophysiological trajectories in Alzheimer's disease progression

**Kiwamu Kudo[1,2]\*[†], Kamalini G Ranasinghe[3†], Hirofumi Morise[1,2], Faatimah Syed[3], Kensuke Sekihara[4], Katherine P Rankin[3], Bruce L Miller[3], Joel H Kramer[3], Gil D Rabinovici[3,5], Keith Vossel[3,6], Heidi E Kirsch[1], Srikantan S Nagarajan[1]\***

[1]Biomagnetic Imaging Laboratory, Department of Radiology and Biomedical Imaging, University of California, San Francisco, San Francisco, United States; [2]Medical Imaging Business Center, Ricoh Company Ltd, Kanazawa, Japan; [3]Memory and Aging Center, UCSF Weill Institute for Neurosciences, University of California, San Francisco, San Francisco, United States; [4]Signal Analysis Inc, Hachioji, Japan; [5]Department of Radiology and Biomedical Imaging, University of California, San Francisco, San Francisco, United States; [6]Mary S. Easton Center for Alzheimer's Research and Care, Department of Neurology, David Geffen School of Medicine, University of California, Los Angeles, Los Angeles, United States

**Abstract** Alzheimer's disease (AD) is characterized by the accumulation of amyloid-β and misfolded tau proteins causing synaptic dysfunction, and progressive neurodegeneration and cognitive decline. Altered neural oscillations have been consistently demonstrated in AD. However, the trajectories of abnormal neural oscillations in AD progression and their relationship to neurodegeneration and cognitive decline are unknown. Here, we deployed robust event-based sequencing models (EBMs) to investigate the trajectories of long-range and local neural synchrony across AD stages, estimated from resting-state magnetoencephalography. The increases in neural synchrony in the delta-theta band and the decreases in the alpha and beta bands showed progressive changes throughout the stages of the EBM. Decreases in alpha and beta band synchrony preceded both neurodegeneration and cognitive decline, indicating that frequency-specific neuronal synchrony abnormalities are early manifestations of AD pathophysiology. The long-range synchrony effects were greater than the local synchrony, indicating a greater sensitivity of connectivity metrics involving multiple regions of the brain. These results demonstrate the evolution of functional neuronal deficits along the sequence of AD progression.

## eLife assessment

This work presents **important** findings for the field of Alzheimer's disease, especially for the electrophysiology subfield, by investigating the temporal evolution of different disease stages typically reported using M/EEG markers of resting-state brain activity. The evidence supporting the conclusions is **convincing** and the methodology as well as the descriptions of the processes are of high quality, although a separation of individuals who are biomarker positive versus negative would have strengthened the results and conclusions of the study.

## Introduction

AD is a neurodegenerative disease characterized by amyloid-β (Aβ) and neurofibrillary tangles of abnormally phosphorylated tau (*DeTure and Dickson, 2019*). Clinical and epidemiological studies have suggested that Aβ accumulation occurs early in the timeline of neuropathological changes in AD,

**\*For correspondence:**
kiwamu.kudo@jp.ricoh.com (KK);
srikantan.nagarajan@ucsf.edu
(SSN)

[†]These authors contributed
equally to this work

**Competing interest:** See page
17

Reviewing Editor: Björn
Herrmann, Baycrest, Canada

likely preceding the accumulation of tau, and subsequent neurodegeneration and cognitive decline (*Jack et al., 2010*; *Sperling et al., 2011*). The neuropathological changes of AD are, therefore, described as a continuum, starting from the presymptomatic stage of proteinopathy and continuing to progress during the symptomatic stage with increasing stages of disease severity (*Sperling et al., 2011*; *Jack et al., 2018*). Transgenic mouse models of AD have shown that AD proteinopathy of Aβ and tau is associated with synaptic and circuit dysfunctions in neural networks (*Busche et al., 2008*; *Ahnaou et al., 2017*; *Busche et al., 2019*). However, the temporal change in synaptic and circuit dysfunction during disease progression in patients with AD remains largely unknown.

Functional deficits in neural networks, especially in the presymptomatic stage, have attracted attention in recent years with the rapidly evolving landscape of plasma biomarkers of early detection and novel therapeutics showing the benefits of early intervention (*Dubois et al., 2016*). In fact, abnormal neural oscillation synchrony has been reported not only in patients along the clinical spectrum of AD, including mild cognitive impairment (MCI) due to AD and AD-dementia (*Jeong, 2004*; *Fernández et al., 2006*; *Stam et al., 2006*; *Koelewijn et al., 2017*; *Nakamura et al., 2018*; *Hughes et al., 2019*; *Ranasinghe et al., 2020*; *Meghdadi et al., 2021*; *Schoonhoven et al., 2022*) but also during the preclinical stages of AD (*Nakamura et al., 2018*; *Ranasinghe et al., 2022a*). Neuronal oscillations observed by noninvasive electrophysiological measures, such as electroencephalography (EEG) and magnetoencephalography (MEG), represent the synchronized activity of excitatory and inhibitory neurons and thus provide sensitive indices of altered neuronal and circuit functions in AD. As synaptic dysfunction is strongly associated with AD proteinopathy, altered neural oscillation synchrony may capture early functional deficits of neural networks even before clinical symptoms appear. However, it remains unknown which neurophysiological signature changes capture such deficits and the temporal evolution of these changes along the timeline of preclinical to MCI to AD dementia stages in clinical populations.

In this study, we investigated the trajectories of neurophysiological changes along the course of clinical progression of AD by examining long-range and local neural synchrony patterns in the resting brain. We hypothesized that frequency-specific long-range and local synchrony abnormalities in neuronal oscillations may precede both neurodegeneration and cognitive deficits. To examine the temporal relationship amongst altered neural synchrony, neurodegeneration, and cognitive deficits, we used data-driven disease progression models, specifically event-based sequencing models (EBM), which have been successfully used to predict AD progression from cross-sectional biomarker data (*Fonteijn et al., 2012*; *Young et al., 2014*; *Young et al., 2018*). In an EBM, disease progression is described as a series of discrete events defined as the occurrence of a particular biomarker reaching a threshold abnormal value, and the estimated likelihood of the temporal sequence of events defines disease progression. Modifying conventional EBMs to find neurophysiological trajectories, we developed a robust EBM framework that is less sensitive to the thresholds for the determination of abnormality thereby resulting in an unbiased estimation of disease stage probability for each study participant.

Leveraging the high spatiotemporal resolution of MEG imaging, we considered two representative neuronal oscillatory synchrony metrics: amplitude-envelope correlation (AEC) and regional spectral power. The AEC and spectral power quantify long-range and local neural synchrony, respectively. Recent test-retest studies of MEG resting-state metrics have revealed that both metrics are highly reliable (*Colclough et al., 2016*; *Wiesman et al., 2022*). To evaluate the frequency specificity of neurophysiological trajectories, three canonical frequency bands, delta-theta (2–7 Hz), alpha (8–12 Hz), and beta (15–29 Hz) bands, were considered. For a metric of global cognitive ability, we used the mini-mental state examination (MMSE) score. Neurodegeneration, which is related to neuronal loss as well as synaptic loss and synapse dysfunction (*Selkoe, 2002*; *Spires-Jones and Hyman, 2014*), is detectable as brain atrophy on structural MRI, and therefore we evaluated neurodegeneration as loss of gray matter (GM) volume, specifically volume loss of the parahippocampal gyrus (PHG), extracted from individual T1 MRIs. We first deployed an Atrophy-Cognition EBM (AC-EBM) with only the neurodegeneration and cognitive decline measures, and then quantitatively examined metrics of long-range and local synchrony of neuronal oscillations corresponding to each estimated disease stage. Next, we deployed two separate Synchrony-Atrophy-Cognition EBMs (SAC-EBMs) which respectively included long-range or local neural synchrony measures along with PHG volume and global cognition, and investigated how the synchrony metrics stratify AD progression. Consistent with our hypothesis,

we found that long-range and local neural synchrony in the alpha and beta bands, but not in the delta-theta band, becomes abnormal at the earliest preclinical stages of AD, preceding both neurodegeneration and cognitive deficits.

## Materials and methods

### Participants

The present study included 78 patients who met National Institute of Aging–Alzheimer's Association (NIA-AA) criteria (*McKhann et al., 2011*; *Albert et al., 2011*; *Jack et al., 2018*) and 70 cognitively-unimpaired older adults. All participants were recruited from research cohorts at UCSF Alzheimer's Disease Research Center (UCSF-ADRC). The diagnosis of AD patients was established by consensus in a multidisciplinary team. Among 78 AD patients, 20 had autopsy-confirmed AD neuropathology, other 41 patients were positive on the Aβ-PET scans, plus another nine patients showed cerebrospinal fluid (CSF) assays of amyloid and tau levels consistent with AD diagnosis. The remaining eight patients were clinically diagnosed, based on clinical evaluations and the characteristic pattern of cortical atrophy on MRI. Control participants were recruited from an ongoing longitudinal study of healthy aging at UCSF-ADRC. The eligibility criteria for cognitively normal controls included normal cognitive performance, normal MRI, and absence of neurological, psychiatric, or other major medical diseases. Forty-seven (out of 70) controls were evaluated with Aβ-PET and 8 were read as positive (39 as negative). The remaining 23 control participants were not evaluated with Aβ-PET. All participants underwent MMSE and a structured caregiver interview to assess the clinical dementia rating scale (CDR). All control participants were identified at CDR 0, indicating cognitively-unimpaired status on the CDR scale. Patients with AD ranged from 0.5 to 2 on the CDR scale. The results of demographic, functional, and cognitive assessments are shown in *Supplementary file 1*. Informed consent was obtained from all participants or their assigned surrogate decision makers. The study was approved by the Institutional Review Board of UCSF (UCSF-IRB 10–02245).

### MRI acquisition and analyses

Structural brain images were acquired using a unified MRI protocol on 3T Siemens MRI scanners (MAGNETOM Prisma or 3T TIM Trio) at the Neuroscience Imaging Center (NIC) at UCSF, within an average of 1.05 years (range: -6.91–0.78) and 0.29 years (range: -2.13–1.29) of the MEG evaluation for controls and patients, respectively. The acquired MRI was used to generate the head model for source reconstructions of MEG sensor data and to evaluate GM volumes. The region-based GM volumes corresponding to the 94 anatomical regions included in the Automated Anatomical Labeling 3 (AAL3) atlas (*Rolls et al., 2020*, *Supplementary file 2*) were evaluated using the Computational Anatomy Toolbox [CAT12 version 12.8.1 (1987)] (*Gaser et al., 2022*), which is an extension of SPM12 (*Penny et al., 2011*); the regional GM volumes were calculated using the morphometry pipeline implemented in CAT12 with default parameters. The total intracranial volume (TIV), the sum of all segments classified as gray and white matter, and CSF, was also calculated for each subject.

### Resting-state MEG

Data acquisition

Each participant underwent 10–60-min resting-state MEG at the UCSF Biomagnetic Imaging Laboratory (BIL). MEG was recorded with a 275-channel full-head CTF Omega 2000 system (CTF MEG International Services LP, Coquitlam, British Columbia, Canada). Three fiducial coils for nasion and left and right preauricular points were placed to localize the position of the head relative to the sensor array and later co-registered with individual MRI to generate an individualized head shape. Data collection was optimized to minimize head movements within the session and to keep it below 0.5 cm. For analysis, a 10-min continuous recording was selected from each subject lying supine and awake with the eyes closed (sampling rate $f_s = 600$ Hz). From the continuous recordings, we further selected a 1-min continuous segment with minimal artifacts (i.e. minimal excessive scatter at signal amplitude) for each subject.

## Pre-processing

Each 1-min sensor signal was digitally filtered using a bandpass filter of 0.5–55 Hz. The power spectral density (PSD) of each sensor signal was computed, and artifacts were confirmed by visual inspections. Channels with excessive noise within individual subjects were removed prior to the next process. When environmental noises larger than a few pT/$\sqrt{\text{Hz}}$ were observed around the 1–5-Hz range in a PSD, the dual signal subspace projection (DSSP) (*Sekihara et al., 2016*) with the lead field vectors computed for each individual subject's head model was applied to the filtered sensor signal for the removal of environmental noise. As a parameter, we chose the dimension of pseudo-signal subspace µ as 50. DSSPs were needed to be applied to 13 of the total 148 subject signals. For the 13 data, the resulting dimension of the spatio-temporal intersection, that is, the degree of freedom to be removed, was 3 or 4. We also applied a preconditioned independent component analysis (ICA) (*Ablin et al., 2018*) to the signal to identify cardiac components and remove them. In each data set, one or two clear cardiac ICA-component waveforms with approximately 1 Hz rhythms were observed, which were easily identified by visual inspections.

## Atlas-based source reconstruction

Isotropic voxels (5 mm) were generated in a brain region of a template MRI, resulting in 15,448 voxels within the brain region. The generated voxels were spatially normalized to individual MRI space, and subject-specific magnetic lead field vectors were computed for each voxel with a single-shell model approximation (*Nolte, 2003*). The voxels for each subject were indexed to 94 cortical/sub-cortical regions included in the AAL3 atlas.

Array-gain scalar beamformer (*Sekihara et al., 2004*) was applied to the 60 s cleaned sensor time series to obtain source-localized brain activity at the voxel level, i.e., voxel-level time courses. Lead field vectors were normalized to avoid the center-of-the-head artifact, and a generalized eigenvalue problem was solved to determine the optimal source orientation (*Sekihara and Nagarajan, 2008*). The beamformer weights were calculated in the time domain; a data covariance matrix was calculated using a whole 60 s time series, and a singular value truncation (threshold of $10^{-6} \times$ maximum singular value) was performed when inverting the covariance matrix. Ninety-four regional time courses were extracted with alignment with the AAL3 atlas by performing a principal component analysis (PCA) across voxel-level time courses within each of the regions and taking a time course of the first principal component. These pre-processing and source reconstructions were performed using in-house MATLAB scripts utilizing Fieldtrip toolbox functions (*Oostenveld et al., 2011*). We also used BrainNet Viewer toolbox (*Xia et al., 2013*) to obtain brain rendering images of regional MEG metrics and GM atrophy.

## MEG resting-state metrics

Based on the regional time courses derived from MEG, we evaluated two measures of neural synchrony: the amplitude-envelope correlation (AEC) and spectral power, which describe long-range and local neural synchrony, respectively. Three canonical frequency bands were considered: delta-theta, alpha, beta bands.

### Amplitude-envelope correlation

The AECs are defined as Pearson's correlation coefficients (PCCs) between any two amplitude envelopes of regional time courses (total $94 \times 93/2 = 4371$ pairs). Regional time courses were first processed by a band-pass filtering, and then their envelopes were extracted by the Hilbert transform. To discount spurious correlations caused by source leakages, we orthogonalized any two band-limited time courses before computing their envelopes by employing a pairwise orthogonalization (*Hipp et al., 2012*; *Sekihara and Nagarajan, 2015*). The AEC with leakage corrections is often expressed as AEC-c and is known as a robust measure (*Briels et al., 2020b*). The pairwise orthogonalization provides asymmetric values between two-time courses; the value depends on which time course is taken as a seed. Therefore, the PCCs between orthogonalized envelopes for both directions were averaged, resulting in a symmetric AEC matrix. Regional AECs, that represent the connectivity strengths of each ROI, were computed by averaging over row/column components of the symmetric AEC matrix.

## Spectral power

The spectral power of a given band, which has often been used as a metric to discriminate patients with AD from controls (*Jeong, 2004*; *Engels et al., 2016*; *Wiesman et al., 2021*), is defined by the ratio of a band power to total power and was calculated from regional PSDs. Regional PSDs were calculated from the 94 regional time courses using Welch's method (50% overlap) with 0.293-Hz (= $f_s$/2048) frequency steps.

## Scalar neural synchrony metrics

To identify general trends in changes in long-range and local synchrony with the severity of AD, we performed group comparisons of the regional synchrony metrics between AD patients and controls. Based on the group contrasts of regional metrics observed, we introduced scalar synchrony metrics by calculating the averages within several regions where large region-level group contrasts were identified. The scalar MEG metrics were used in the SAC-EBMs.

## Metric trajectory analyses

### Event-based sequencing modeling

Imaging and neuropsychological biomarkers for AD are continuous quantities taking values from normal to severe, while the stages of the disease are discrete and are identified by estimating the values of biomarkers (*Sperling et al., 2011*). As a data-driven disease progression model, an event-based sequencing model (EBM) has been proposed that allows us to make inferences about disease progression from cross-sectional data (*Fonteijn et al., 2012*; *Young et al., 2014*; *Young et al., 2018*). In an EBM, disease progression is described as a series of metric events, where events are defined as the occurrences of abnormal values of metrics, and the values of events act as thresholds to determine discrete stages of disease (*Fonteijn et al., 2012*). The model infers temporal sequences of the events from cross-sectional data.

It is also possible to set multiple events per metric by defining them as occurrences of taking certain $z$-scores within the range from initial to final $z$-scores ([$z_{initial}$ $z_{final}$]), in which $z$-scores for each metric linearly increase between all consecutive events and the stages are located at temporal midpoints between the two consecutive event occurrence times (*Young et al., 2018*). In this linear $z$-score event model, a metric trajectory is described as a series of metric values evaluated at estimated stages.

We developed a robust EBM framework to quantify metric trajectories on the basis of the linear $z$-score model, employing the following form of a data likelihood:

$$P(Z|S) = \prod_{j=1}^{J} \sum_{k=1}^{N+1} p(t_j = k)p(Z_j|S, t_j = k), \tag{1}$$

where $N$ denotes a total number of events, $S$ denotes a sequence of the events, and $i$, $j$, and $k$ are the indices of metric, subject, and stage, respectively. $J$ is the number of subjects ($J = 148$). $I$ is the number of metrics: $I = 2$ for an AC-EBM and $I = 3$ for an SAC-EBM, respectively. The symbol $t_j$ denotes stages for each subject $j$, and a conditional probability, $p(Z_j|S, t_j = k)$, describes the probability that a subject $j$ takes biomarker values of $Z_j$ given a sequence of events $S$ and that $t_j = k$ (that is, the subject $j$ is in a stage $k$). The symbol $Z_j = [z_{1j}, z_{2j}, \ldots, z_{Ij}]^T$, where $z_{ij}$ denotes the $z$-score of a metric $i$ for a subject $j$, and the symbol $Z = [Z_1, Z_2, \ldots, Z_J]$ describing the data matrix with the $I \times J$ dimension. Since there are $N + 2$ event occurrence times including initial and final times, $N + 1$ stages are provided. When employing *Equation 1*, we assumed that the prior distribution in which the subject $j$ is in a stage $k$ is uniform: $p(t_j = k) = (N + 1)^{-1}$. We also assumed that the prior probability of $p(Z_j|S, t_j = k)$ arises from independent Gaussian distributions for each metric $i$, resulting in a multivariate factorized prior. Hence,

$$p(Z_j|S, t_j = k) \propto \prod_{i=1}^{I} \exp\left(-\frac{(z_{ij} - \mu_i(k))^2}{2}\right). \tag{2}$$

The symbol $\mu_i(k)$ denotes a value of the $z$-score of a metric $i$ at a stage $k$ and is given by a linearly interpolated midpoint $z$-score between two $z$-scores evaluated at consecutive event occurrence times. The

goal of this formulation is to evaluate the posterior distribution that a subject $j$ belongs to a stage $k$, $p(t_j = k|Z_j, \bar{S})$, with the most likely order of events $\bar{S}$.

The most likely order of the events is given by the sequence of events, $S$, which maximizes the posterior distribution $P(S|Z) = P(S)P(Z|S)/P(Z)$. Under the assumption that the prior $P(S)$ is uniformly distributed (*Fonteijn et al., 2012*), the most likely sequence is obtained by solving the maximum likelihood problem of maximizing *Equation 1*. To solve the problem, for a given set of events, we performed Markov chain Monte Carlo (MCMC) sampling on sequences and chose the maximum likelihood sequence from 50,000 MCMC samples. In the generation of the MCMC samples, we initialized the MCMC algorithm with an initial sequence close to or equal to the maximum likelihood solution by running an ascent algorithm 10 times from different initialization points, i.e., randomly generated event sequences (*Fonteijn et al., 2012*).

### z-scoring of metrics

We computed $z$-scores of the PHG volume, MMSE score, and scalar neural synchrony metrics to utilize them in the EBM frameworks. Since a linear $z$-score model assumes a monotonous increase in $z$-scored metrics along disease progression (i.e. higher stage denotes more severity), 'sign-inverted' $z$-scores were introduced to the metrics with decreasing trends along disease progression. Specifically, for GM volumes, MMSE score, and neural synchrony metrics in the alpha and beta bands, the $z$-score of a metric $i$ for a subject $j$ was defined by $z_{ij} = (\bar{x}_i^C - x_{ij})/\sigma_i^C$, where $x_{ij}$ denotes a value of a metric $i$ for a subject $j$, and $\bar{x}_i^C$ and $\sigma_i^C$ denote the mean and standard deviation (SD) of the metric values of the controls, respectively. For the delta-theta-band neural synchrony metrics, $z$-scores were defined in a standard way as $z_{ij} = (x_{ij} - \bar{x}_i^C)/\sigma_i^C$. Using these $z$-scored metrics, the initial and final events, $z_{\text{initial}}$ and $z_{\text{final}}$, for each metric were set as the bottom and top 10% average $z$-scores, respectively.

### Events-setting optimization

In addition to the initial and final events of the $z$-score, $z_{\text{initial}}$ and $z_{\text{final}}$, we set three events for each metric because various possible curves of the metric trajectories were supposed to be well expressed by three variable points with two fixed points. For example, in an AC-EBM analysis, that is, a two-metric trajectory analysis for PHG volume loss and MMSE decline, a total of six events were considered ($N = 6$). The metric trajectory as a series of stage values $\mu_i(k)$ is sensitive to event settings because predefined events do not necessarily capture appropriate boundaries between disease stages. To determine disease stages less sensitive to specifications of the $z$-score events, we tried several sets of events and selected the set of events with the highest data likelihood among the trials. Specifically, we searched for the set of events that better fits the data $Z$ by trying all combinations of three $z$-scores from $\{0.2, 0.3, 0.4, 0.5, 0.6, 0.7, 0.8\}$-quantiles for each metric. The number of combinations of events for each metric was accordingly ($_7C_3 =$)35. Therefore, MCMC samplings (50,000 samples for each set of events) were performed 1225 times for an AC-EBM and 42,875 times for an SAC-EBM, respectively, to find the set of events and their sequence $\bar{S}$ with the highest data likelihood. This exhaustive search for optimal event settings, which was not implemented in a conventional linear $z$-score EBM (*Young et al., 2018*), is shown schematically in *Figure 1—figure supplement 1*.

Although it is tractable to directly evaluate $P(Z|S)$ for all ordered arrangements of $z$-score events when the number of the permutations for each set of $z$-score events is just $20(=_6 C_3)$ for $I = 2$ (*Supplementary file 3*) and $1680(=_9 C_3 \times_6 C_3)$ for $I = 3$, such a direct evaluation is not tractable when $I > 3$ and requires MCMC sampling. Furthermore, MCMC enables the computation of sequence statistics. Therefore, we used MCMC sampling from which we could compute the positional variance estimates for each event (see *Figure 1—figure supplement 1* and *Figure 1—figure supplement 2*).

## Trajectory computations

Given the most likely sequence $\bar{S}$ as a result of the exhaustive search, the probabilities that a subject $j$ falls into a stage $k$ are evaluated by the posterior distribution:

$$p_j(k) \equiv p(t_j = k|Z_j, \bar{S}) = \frac{p(Z_j|\bar{S}, t_j = k)}{\sum_{k'} p(Z_j|\bar{S}, t_j = k')}. \tag{3}$$

These probabilities describe the contribution of a subject $j$ to stage $k$, allowing us to evaluate the stage value of any metric $x$ of $i$ at a stage $k$ as a weighted mean:

$$\bar{x}_i(k) = \frac{\sum_{j=1}^{J} p_j(k) \cdot x_{ij}}{\sum_{j=1}^{J} p_j(k)}. \tag{4}$$

Then, we represented the trajectory of the metric $i$ by a series of the stage values, $\bar{x}_i(k)$. The standard error (SE) of the weighted mean at stage $k$ was evaluated by

$$\mathrm{SE}(k) = \sigma_i \cdot \sqrt{\frac{\sum_{j=1}^{J} p_j(k)^2}{\left(\sum_{j=1}^{J} p_j(k)\right)^2}}, \tag{5}$$

where $\sigma_i$ is a standard deviation of a metric $i$. This definition of SE provides an usual expression of the standard error of the mean, $\sigma_i/\sqrt{J}$, if all subjects contributed equally to all stages.

These formulations of trajectories were applied to several metrics. In the AC-EBM, the metrics $i$ denote the PHG volume loss $z$-score and the MMSE scores. In the SAC-EBM, they denote each scalar neural synchrony metric in addition to the PHG volume loss $z$-score and the MMSE score. We also used *Equation 4* to evaluate the progressions of the regional neural synchrony metrics and the regional GM volume loss $z$-scores along the estimated EBM stages. When evaluating the proportion of subjects classified into each stage, we treated $(x_{i1}, x_{i2}, \ldots, x_{iJ})$ as a vector in which a metric $i$ represents a category of subjects provided by the CDR scale. For example, when evaluating the ratio of subject with CDR 0.5, $x_{ij} = 1$ only when a subject $j$ has CDR scale of 0.5, otherwise $x_{ij} = 0$.

## Statistical analyses

To test demographic differences between AD patients and controls, the unpaired $t$-test was used for age and the chi-square test for sex. The age was defined at the time of the MEG scan date. In statistical analyses, p-values below 0.05 were considered statistically significant. For group comparisons of GM volumes, MMSE scores, and neural synchrony metrics, two-sided significance tests (against a null value of zero) were performed using the general linear model (GLM). For statistical tests on GM volumes, TIV, age, and the difference between MRI and MEG dates were included as covariates. For statistical tests on MMSE scores, age and the difference between MMSE and MEG dates were included as covariates. For statistical tests of neural synchrony metrics, age was included as a covariate. The problem of multiple comparisons between 94 regions was solved by controlling the Benjamini-Hochberg false discovery rate (FDR) (*Benjamini and Hochberg, 1995*). The FDR-adjusted $p$-value (i.e., $q$-value) below 0.05 or 001 was considered statistically significant.

A non-parametric test was performed to statistically compare metrics between stages, i.e., to test statistical significance of the difference between stage values represented by weighted means [e.g. stage $k$ vs $k'$: $\delta x = \bar{x}_i(k) - \bar{x}_i(k')$]. For a metric $i$, we used bootstrap resampling (50,000 samples) of an original data set, $\boldsymbol{x}_i = (x_{i1}, x_{i2}, \ldots, x_{iJ})$, to generate new data sets, $(x_{i1}^{\star}, x_{i2}^{\star}, \ldots, x_{iJ}^{\star})$, using a random number generator, where each $x_{ij}^{\star}$ is one of the components of the original data set $\boldsymbol{x}_i$. We then calculated the weighted means $\bar{x}_i^{\star}(k)$ (*Equation 4*) for each sample. The same procedures were performed for stage $k'$, obtaining weighted means $\bar{x}_i^{\star}(k')$ for each sample. We then tested the null hypothesis that a weighted mean in stage $k$, $\bar{x}_i(k)$, is equal to a weighted mean in stage $k'$, $\bar{x}_i(k')$, evaluating the null distribution of differences in the weighted mean values, $\delta x^{\star} = \bar{x}_i^{\star}(k) - \bar{x}_i^{\star}(k')$. The problem of multiple comparisons across stages was solved by controlling the FDR. The $q$-value below 005 was considered statistically significant.

## Results

### Participant demographics

This study included a cohort of 78 patients with AD (50 female; 28 male) including 35 patients with AD dementia and 43 patients with MCI due to AD, and also included 70 cognitively-unimpaired older adults as controls (41 female; 29 male). The CDR scales were 0 for the cognitively-unimpaired controls, 0.5 for patients with MCI, and 1 ($n = 27$) or 2 ($n = 8$) for patients with AD dementia. There were no differences in sex distribution between the AD and control groups [$\chi^2(1) = 0.477; p = 0.49$].

The average age at the time of MEG was slightly higher in the control group than patients with AD (controls, mean ± SE: $70.5 \pm 0.99$, range: 49.5–87.7; AD, mean ± SE: $63.9 \pm 1.01$, range: 49.0–84.4) [unpaired $t$-test: $t(146) = -4.708$; $p < 0.001$]. The mean MMSE in patients with AD was $22.7 \pm 0.43$ (mean ± SE) while the mean MMSE in the controls $29.2 \pm 0.48$. MMSE scores were adjusted for age and time differences between MMSE administration and MEG scan using a GLM (*Figure 1—figure supplement 3B*). MMSE-decline $z$-scores, $z_{\mathrm{MMSE}}$, were standardized by adjusted MMSE scores of the control group and sign-inverted (*Figure 1B*).

Group comparisons of GM volumes for each of the anatomical regions included in the AAL3 atlas showed that GM volumes in the temporal regions are significantly smaller in AD patients than in controls (*Figure 1—figure supplement 4*; *Supplementary file 4*). Among temporal GM volumes, we focused on a volume of PHG as a key indicator of neurodegeneration in AD progression. The PHG includes the perirhinal and entorhinal cortices of the medial temporal lobe (MTL), and MRI-based studies have reported that MTL volume decreases, especially in the perirhinal and entorhinal cortices, in the early stages of typical AD (*Teipel et al., 2006*; *Echávarri et al., 2011*; *Matsuda, 2016*). In this study, PHG volume was defined as a sum of the volumes of left- and right-hemisphere PHGs. The average volume of PHG in AD patients ($7.99\,\mathrm{ml} \pm 0.09$) was significantly lower than in controls ($9.28\,\mathrm{ml} \pm 0.11$) [unpaired $t$-test: t(143) = -9.508; ***$p$ < 0.001] (*Figure 1—figure supplement 3A*); the PHG volumes were adjusted for TIV, age, and the difference between MRI and MEG dates by including them in a GLM as covariates. PHG volume loss $z$-scores, $z_{\mathrm{PHG}}$, were standardized by the adjusted PHG volumes of the control group and sign-inverted (*Figure 1A*).

## Abnormal frequency-specific long-range and local neural synchrony in AD

We performed group comparisons of MEG metrics. Three canonical frequency bands were considered: $2 - 7\,\mathrm{Hz}$ (delta-theta), $8 - 12\,\mathrm{Hz}$ (alpha), and $15 - 29\,\mathrm{Hz}$ (beta) bands (*Figure 2—figure supplement 1*). For regional long-range synchrony (AEC), increases in delta-theta-band synchrony in patients with AD were identified in frontal regions, and reductions in alpha- and beta-band synchrony were identified in the whole brain (*Figure 2—figure supplement 2C, E*; *Supplementary file 5*). These regional contrasts were similar to those observed between AD dementia and subjective cognitive decline (SCD) in MEG/EEG studies (*Schoonhoven et al., 2022*; *Briels et al., 2020a*). For regional local synchrony (spectral power), increases in delta-theta-band power in patients with AD were identified in the whole brain, and reductions in alpha- and beta-band power were identified in temporal regions and the whole brain, respectively (*Figure 2—figure supplement 2D, F*; *Supplementary file 6*). These regional contrasts were similar to those observed between MCI and controls in a multicenter study of MEG (*Hughes et al., 2019*).

Based on the group contrasts of regional metrics observed, we introduced six scalar metrics to quantify long-range and local synchrony: [i] frontal delta-theta-band AEC, [ii] whole-brain alpha-band AEC, [iii] whole-brain beta-band AEC, [iv] whole-brain delta-theta-band spectral power, [v] temporal alpha-band spectral power, and [vi] whole-brain beta-band spectral power. We computed the average within several regions where large group contrasts were identified at the region level (the temporal and frontal regions of interest (ROI) are illustrated in *Figure 2—figure supplement 3*). Consistent with regional group comparisons, the long-range and local synchrony scalar metrics in delta-theta band increased in AD patients compared to controls, and the long-range and local synchrony scalar metrics in alpha and beta bands were reduced in AD patients compared to controls (*Figure 2—figure supplement 2A, B*). We also calculated the $z$-scores, $z_{\mathrm{MEG}}$, of each scalar metric that was used in the SAC-EBMs.

## PHG volume loss precedes the MMSE decline in AD progression

An AC-EBM analysis with the two metrics, PHG volume loss, $z_{\mathrm{PHG}}$, and MMSE decline, $z_{\mathrm{MMSE}}$, was performed for six events ($N = 6$; three events for each metric). Robust event thresholds were determined by the exhaustive search of multiple event thresholds ($z$-score thresholds) and choosing the set of event thresholds that maximize the data likelihood (*Equation 1*). The AC-EBM provided seven stages, each located between consecutive event occurrence times. The resulting posterior probabilities, $p_j(k)$, that a subject $j$ belongs to a stage $k$ are shown in *Figure 1C*. Based on the probabilities, the ratio of subjects classified into each stage was calculated as the probability-based weighted mean

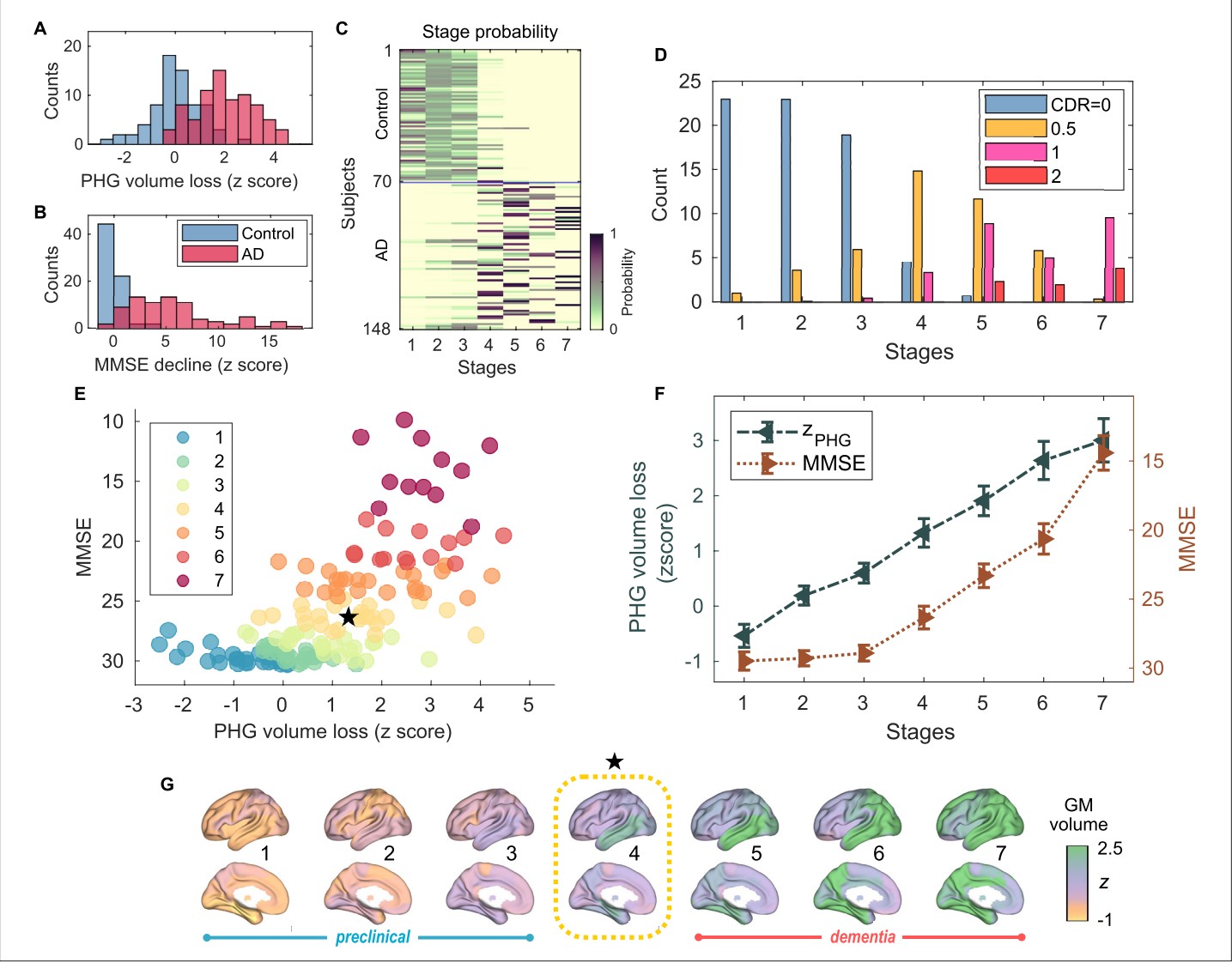

**Figure 1.** Atrophy-cognition event-based sequencing model (EBM) staging of Alzheimer's disease (AD) progression. (**A**) Histogram of parahippocampal gyrus (PHG) volume loss $z$-scores, $z_{\mathrm{PHG}}$. (**B**) Histogram of mini-mental state examination (MMSE)-decline $z$-scores, $z_{\mathrm{MMSE}}$. The $z$-scores for PHG volume loss and MMSE were standardized by the adjusted scores of the control group and sign-inverted so that higher $z$-scores denote more severity. (**C**) Posterior probabilities, $p_j(k)$, that a subject $j$ belongs to a stage $k$ evaluated by the Atrophy-Cognition EBM (AC-EBM). (**D**) The ratio of subjects classified to each stage; blue: Control (clinical dementia rating, CDR 0), orange: MCI due to AD (CDR 0.5), pink: mild AD dementia (CDR 1), and red: moderate AD dementia (CDR 2). (**E**) Distribution of the stages in the space spanned by PHG volume loss and MMSE score. Each subject $j$ was distinctly assigned to one of the stages with the highest posterior probability, $\mathrm{argmax}_k p_j(k)$. The colors of the dots denote the seven stages. A star symbol denotes the probability-based weighted means of $z_{\mathrm{PHG}}$ and MMSE scores at stage 4 and MMSE = 26.3 ($\pm 0.82$). The values in parentheses denote the standard error (SE; *Equation 5*) of the weighted means. (**F**) Trajectories of PHG volume loss and MMSE score as a function of the seven stages. Probability-based weighted means ($\pm$ SE) are shown. The initial and final $z$-scores used in the AC-EBM were: ($z_{\mathrm{initial}}, z_{\mathrm{final}}$) = ($-1.372, 3.804$) for PHG volume loss and ($-0.902, 12.712$) for MMSE decline, respectively. (**G**) Progression of GM volume loss ($z$-scores) from stage 1–7. Regional gray matter (GM) atrophy in the predicted stage of mild cognitive impairment (MCI) (stage 4) was circled with a dotted line.

The online version of this article includes the following figure supplement(s) for figure 1:

**Figure supplement 1.** Steps for computing metric trajectories.

**Figure supplement 2.** An example of positional variance diagram of $z$-score events.

**Figure supplement 3.** Group comparisons of parahippocampal gyrus (PHG) volumes and mini-mental state examination (MMSE) scores.

**Figure supplement 4.** Group comparison of gray matter (GM) volumes.

(*Figure 1D*). The ratio of subjects with CDR 05 was highest in stage 4, and the ratio of controls with CDR 0 in stage 4 was small compared to those in less severe stages of 1–3, indicating that stage 4 corresponds best to clinical MCI due to AD.

The trajectory of PHG volume loss preceded that of MMSE decline (*Figure 1F*), consistent with the relationship between brain atrophy and cognitive decline described in a hypothetical model of biomarker trajectories (*Jack et al., 2010*; *Sperling et al., 2011*). *Figure 1E* visualizes the distribution of the seven stages in the PHG volume loss versus MMSE score. At stage 4, the value of $z_{PHG}$ of $1.33 \pm 0.258$ was in the range of 1–2. This $z$-score range of PHG volume loss corresponds to a mild-atrophy range representing approximately the MCI stage, e.g., in the voxel-based specific regional analysis system for AD (VSRAD) software (*Hirata et al., 2005*; *Matsuda et al., 2012*). Furthermore, the MMSE score of $26.3 \pm 0.82$ at stage 4 was in the range of 23–27. This range of MMSE scores is considered typical for MCI due to AD (*Tsoi et al., 2015*). Stage 4, therefore, corresponds to MCI stage, whereas stages 3 and 5 correspond to preclinical-AD and mild AD-dementia stages, respectively.

The GM volume $z$-scores as a function of the seven stages showed that prominent atrophy with $z > 1$ is observed in the temporal regions starting at stage 4 (*Figure 1G*). This trajectory of GM volume approximated the evolution of brain atrophy in the typical progression of AD reported in MRI-based studies; GM volume loss in AD starts in the MTL in the MCI stage, spreads to the lateral temporal and parietal lobes in the mild AD-dementia stage, and spreads further to the frontal lobe in moderate AD-dementia (*Scahill et al., 2002*; *Tondelli et al., 2012*; *Jack et al., 2013*).

These results of the AC-EBM indicate that the PHG volume loss precedes the MMSE decline, and their metric changes track the stages of AD from preclinical AD to moderate AD-dementia. The order of events for GM volume loss and cognitive decline was consistent with the observation that cognitive decline in the early stage of AD progression reflects neuronal loss in the medial temporal regions (*Jack et al., 2018*; *DeTure and Dickson, 2019*).

## Neural synchrony progressively changes throughout the AD stages estimated by AC-EBM

For the seven stages determined by the AC-EBM (*Figure 1E–G*), long-range and local neural synchrony profiles were estimated (*Figure 2*). Along the EBM stages, the delta-theta-band synchrony was consistently increased and the alpha and beta-band synchrony was consistently decreased. Neural synchrony showed prominent changes around stage 4 (clinical stage of MCI due to AD). The long-range synchrony in the alpha and beta bands decreased steadily in stages 1–3 and then decreased further in stage 4 (*Figure 2A*). Local synchrony in the beta band also decreased by half from 1 to 4 (*Figure 2B*). On the contrary, there were little changes in delta-theta-band long-range synchrony and delta-theta- and alpha-band local synchrony from stage 1 to 3 but these changes became prominent after stage 3.

Regional patterns of long-range and local synchrony as a function of the seven stages indicated that prominent changes manifest themselves at stage 4 (*Figure 2E and F*; *Supplementary file 7* and *Supplementary file 8*). The regions with prominent deviations overlapped with the regions where a significant increase and decrease in neural synchrony was observed in the group comparisons (*Figure 2—figure supplement 2E, F*).

The changes in neural synchrony metrics with AD progression indicate that neural synchrony is a sensitive indicator of functional change along AD progression. To further investigate the temporal association of functional deficits with neurodegeneration and cognitive decline, we included neural synchrony in addition to the PHG volume loss and MMSE decline in the EBM frameworks, performing SAC-EBMs.

## Long-range synchrony changes in the alpha and beta bands precede PHG volume loss and MMSE decline

SAC-EBMs that include PHG volume loss, $z_{PHG}$, MMSE decline, $z_{MMSE}$, and long-range synchrony metric $z$-scores, $z_{MEG}$, were performed setting a total of nine events ($N = 9$). SAC-EBMs separately included long-range neural synchrony metrics in the delta-theta, alpha, and beta bands. Each EBM determined the order of nine events, thus defining ten stages (*Figure 3—figure supplement 1*; for the corresponding positional variance diagrams of the optimal set of $z$-score events in the SAC-EBMs,

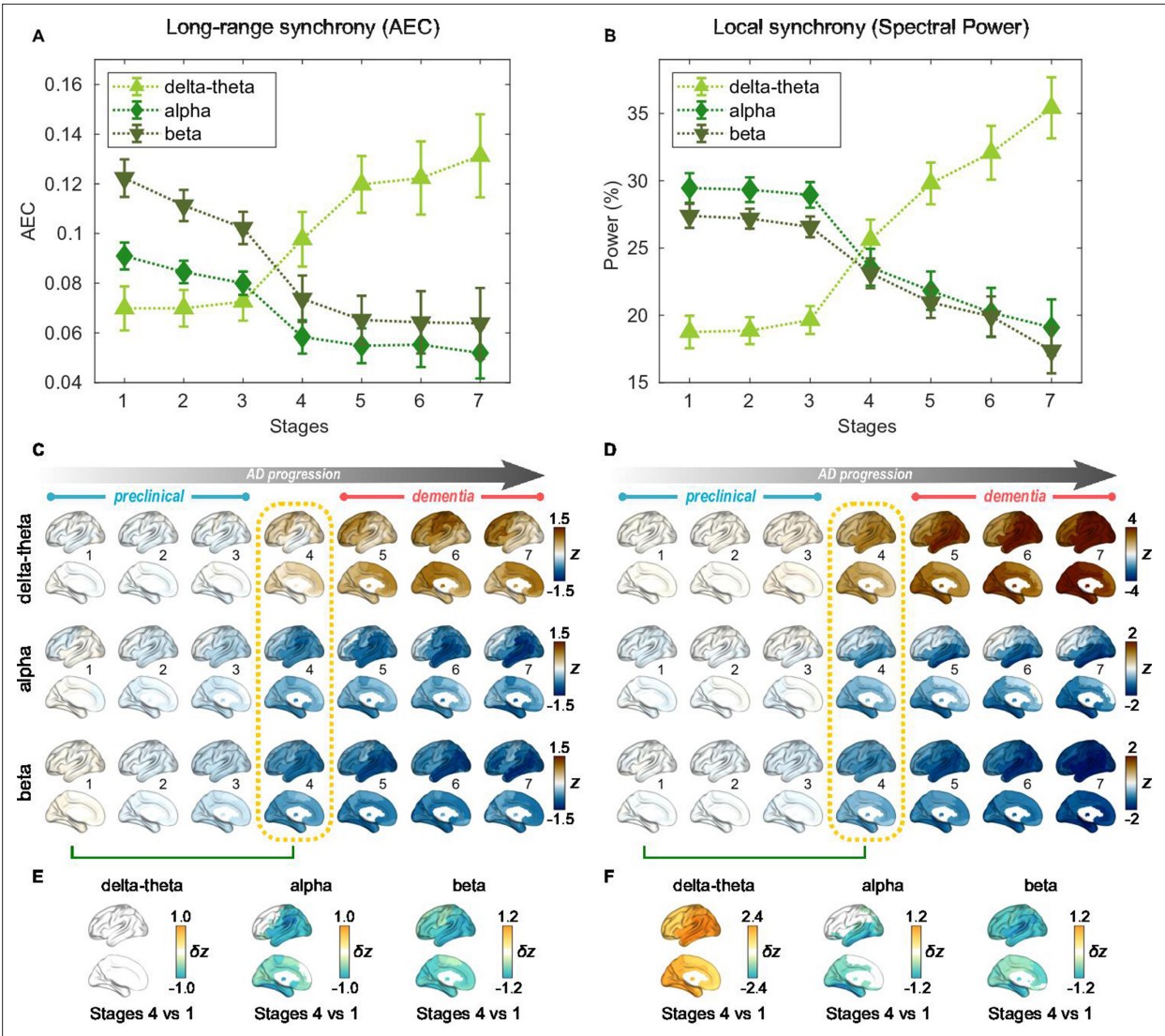

**Figure 2.** Profiles of neural synchrony as a function of the Alzheimer's disease (AD) stages estimated by Atrophy-Cognition EBM (AC-EBM). (**A,B**) Profiles of amplitude-envelope correlation (AEC) (**A**) and spectral power (**B**) as a function of the seven stages, showing probability-based weighted means (± SE). Neural synchrony increased monotonously with AD progression in the delta-theta band and decreased monotonously in the alpha and beta bands. (**C,D**) Regional AEC (**C**) and spectral power (**D**) as a function of the seven stages. Deviations from the neural-synchrony spatial patterns averaged over the controls are displayed. The deviations were evaluated using the probability-based weighted means of $z$-scores standardized by the controls. Spatial patterns in the mild cognitive impairment (MCI) stage (stage 4) were circled with dotted lines. (**E,F**) Changes in neural synchrony during the preclinical stages. Regional comparisons between two stages (stages 4 vs 1) are shown based on non-parametric tests of weighted mean differences $\delta z$. Differences that exceed the threshold ($q < 0.05$) are displayed. There were no significant differences in long-range synchrony in the delta-theta band.

The online version of this article includes the following figure supplement(s) for figure 2:

**Figure supplement 1.** Normalized power spectral densities (PSDs) for the Alzheimer's disease (AD) and control groups.

**Figure supplement 2.** Group comparisons of magnetoencephalography (MEG) metrics.

**Figure supplement 3.** Frontal and temporal regions of interest.

see *Figure 3—figure supplement 2* and *Figure 3—figure supplement 3*). The resulting posterior probabilities, $p_j(k)$, that a subject $j$ belongs to a stage $k$ are shown in *Figure 3—figure supplement 4*.

For all frequency bands, around stages 5 and 6, the weighted means of PHG volume loss $z$-scores were in the range of 1–2 and the MMSE scores were in the range of 23–27 (*Figure 3B, F and J*). Furthermore, the ratio of subjects with CDR 05 was high around stage 5 (*Figure 3A, E1*). These indicated that stage 5 best represents the onset of clinical MCI stage, and stages 1-4, where MMSE scores remain almost constant at or near 30, correspond to the preclinical stages of AD. Changes in

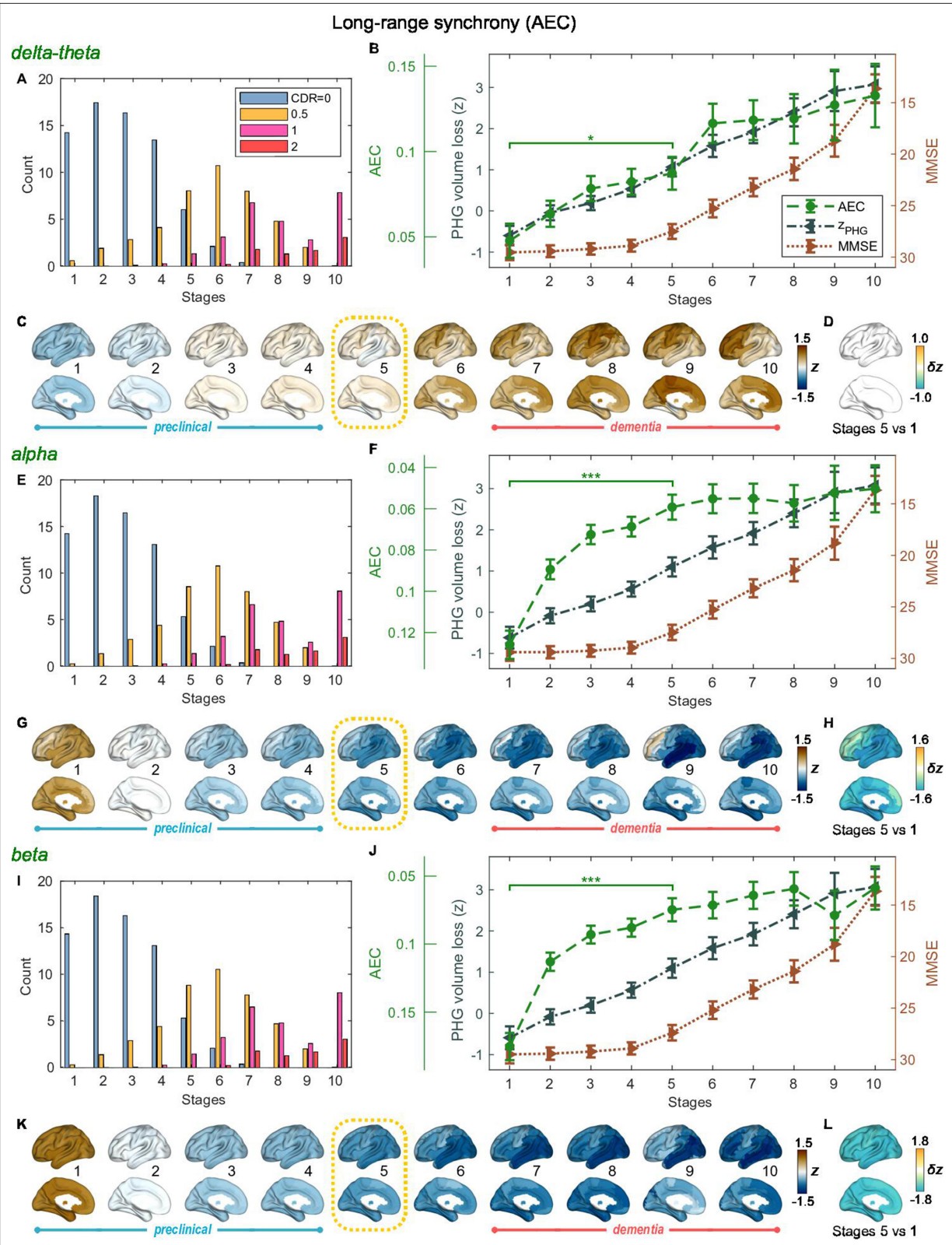

**Figure 3.** Trajectories of long-range neural synchrony in delta-theta, alpha, and beta-bands from SAC-EBMs. (**A, E, I**) The ratio of subjects classified to each stage. The ratio was evaluated on the basis of the probabilities that each subject will be assigned to each of the ten stages. (**B, F, J**) Trajectories of long-range synchrony, parahippocampal gyrus (PHG) volume loss, and mini-mental state examination (MMSE) score as a function of the ten stages, showing probability-based weighted means (± SE). The asterisks (*q < 0.05 and ***q < 0.001, false discovery rate - FDR corrected) denote statistical

*Figure 3 continued on next page*

*Figure 3 continued*

significance in comparisons between stages 5 vs 1. All stage pairs with significant weighted mean differences are listed in **Supplementary file 9**. Initial and final $z$-scores of long-range synchrony used in the SAC-EBMs were: $(z_{\text{initial}}, z_{\text{final}}) = (-1.083, 2.811), (-1.542, 1.605)$, and $(-1.624, 1.641)$ in the delta-theta, alpha, and beta bands, respectively. (**C, G, K**) Regional amplitude-envelope correlation (AEC) along the stages. The deviations from the regional patterns of the control group are shown. The regional patterns at the onset of the mild cognitive impairment (MCI) stage were circled with dotted lines. (**D,H,L**) Changes in regional patterns during the preclinical stages. Regional comparisons between two stages are shown based on non-parametric tests of weighted mean differences $\delta z$. Differences exceeding threshold ($q < 0.05$, FDR corrected) are displayed. The top 10 regions with significant differences are listed in **Supplementary file 10**.

The online version of this article includes the following figure supplement(s) for figure 3:

**Figure supplement 1.** Event sequences and trajectories determined by SAC-EBMs.

**Figure supplement 2.** Positional variance diagrams of the $z$-score events in SAC-EBMs.

**Figure supplement 3.** Markov chain Monte Carlo (MCMC) samples of the sequence of the optimal set of $z$-score events in the SAC-EBM including alpha-band amplitude-envelope correlation (AEC).

**Figure supplement 4.** Posterior probabilities evaluated by the SAC-EBMs.

long-range synchrony during the preclinical stages are shown as statistical bars, and the region-level changes are shown in *Figure 3C–D, G–H, and K–L*.

Long-range synchrony in the alpha and beta bands decreased markedly during the preclinical stages of AD, preceding both PHG volume loss and MMSE decline. Specifically, between stages 1 and 4, the alpha- and beta-band long-range synchrony decreased by more than 80% of the total drop seen from stage 1 to 10. The whole brain, but especially the temporal area, was involved in these prominent preclinical changes (*Figure 3H and L*). In contrast, the trajectory of delta-theta-band long-range synchrony (*Figure 3B and C*) was almost identical to the evolution of the PHG volume loss throughout the stages, but a large variation occurred around the MCI stages (stages 5 and 6) as was found in the AC-EBM (*Figure 2A*). There were no significant increases in region-level synchrony in delta-theta band during the preclinical stages (*Figure 3D*), consistent with an observation seen in the AC-EBM (*Figure 2F*).

The trajectory shapes of the PHG volume loss (almost linear) and MMSE scores (half parabola) were similar to those obtained in the AC-EBM (*Figure 1G*). This indicates that prominent changes in alpha- and beta-band long-range synchrony during preclinical stages can be utilized to stratify the preclinical stages determined only by neurodegeneration and cognitive deficits.

## Local synchrony changes in the alpha and beta bands precede PHG volume loss and MMSE decline

SAC-EBMs including PHG volume loss, MMSE decline, and local synchrony metric $z$-scores were performed, separately considering delta-theta-, alpha-, and beta-band local synchrony metrics. When considering delta-theta and alpha bands, around stages 6 and 7, the PHG volume loss $z$-scores were in the range of 1–2 and the MMSE scores were in the range of 23–27 (*Figure 4B and F*), indicating that stage 6 best represents the onset of the MCI stage. Furthermore, the ratios of subjects with CDR 0.5 were high in stages 6 and 7 (*Figure 4A and E*). For the beta band, based on similar observations, stage 6 best represented the MCI stage (*Figure 4I and J*). For all frequency bands, stages 1-5, where MMSE scores remain almost constant at or near 30, corresponded to the preclinical stages of AD. The changes in local synchrony during the preclinical stages are shown as statistical bars, and the corresponding region-level changes are shown in *Figures 4C–D, G–H and K–L*.

Local synchrony in the alpha and beta bands decreased during the preclinical stages of AD, preceding both PHG volume loss and MMSE decline (*Figure 4F and G* and *Figure 4J and K*). On the contrary, the local synchrony in the delta-theta band increased, lagging the evolution of PHG volume loss (*Figure 4B and C*). Specifically, the alpha-band local synchrony decreased considerably by the onset of the MCI stage, showing significant reductions in the temporal regions (*Figure 4H*) during the preclinical stages (stages 6 vs 1). It is noted that these trends were inconsistent with those found in the AC-EBM (*Figure 2B*), especially within the preclinical stages, where there was little change found in the alpha-band local synchrony. This can be interpreted as evidence that early stages in AD progression may be better characterized by including neurophysiological markers as AD indicators. Beta-band local synchrony also decreased during the preclinical stages, preceding PHG volume loss

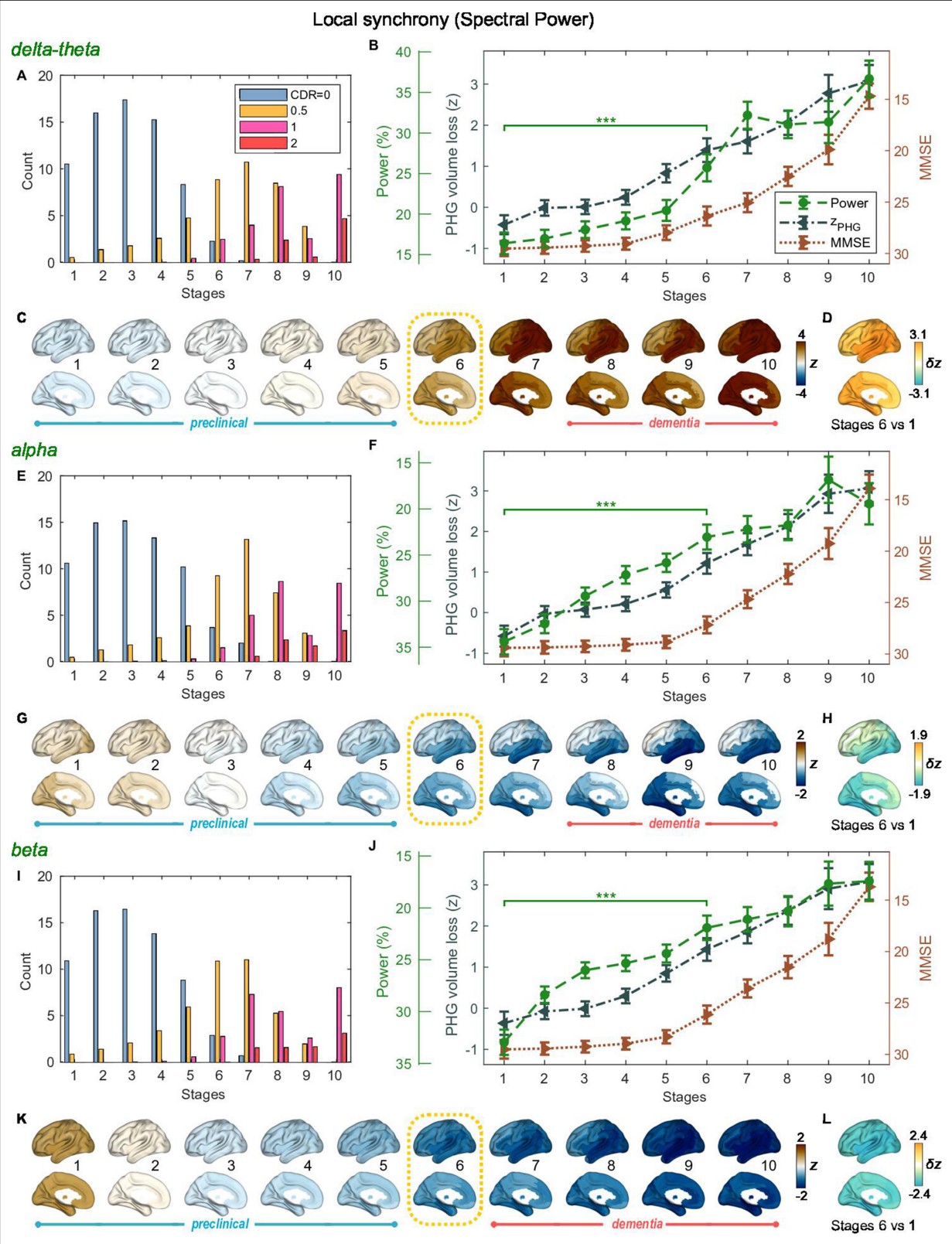

**Figure 4.** Trajectories of local neural synchrony in delta-theta, alpha, and beta bands from SAC-EBMs. (**A, E, I**) The ratio of subjects classified to each stage. (**B, F, J**) Trajectories of local synchrony, parahippocampal gyrus (PHG) volume loss, and mini-mental state examination (MMSE) score as a function of the 10 stages, show the weighted mean (± SE). Asterisks (***q < 0.001, FDR corrected) denote statistical significance in comparisons between stages 6 vs 1. All stage pairs with significant weighted mean differences are listed in **Supplementary file 11**. The initial and final z-scores of

*Figure 4 continued on next page*

*Figure 4 continued*

local synchrony used in the SAC-EBMs were: $(z_{initial}, z_{final}) = (-1.329, 6.097)$, $(-1.461, 2.866)$, and $(-1.810, 2.784)$ in the delta-theta, alpha, and beta bands, respectively. (**C, G, K**) Regional spectral power along the stages. Deviations from the regional patterns of the control group are shown. The regional patterns at the onset of the mild cognitive impairment (MCI) stages were circled with dotted lines. (**D, H, L**) Changes in regional patterns during the preclinical stages. Regional comparisons between two stages are shown based on nonparametric tests of weighted mean differences $\delta z$. Differences exceeding threshold ($q < 0.05$, false discovery rate - FDR corrected) are displayed. The top 10 regions with significant differences are listed in **Supplementary file 12**.

and MMSE decline; by stage 5, the beta-band power decreased by approximately 55% of the total drop seen throughout the stages, and the reductions were observed in the whole brain (**Figure 4L**). Unlike the local synchrony trajectories in the alpha and beta bands, the local synchrony in the delta-theta band increased. The *hyper*-synchrony lagged the evolution of the loss of PHG volume in the preclinical stages and made a large jump around the stages 6 and 7 (**Figure 4D**).

As shown in the previous section, large alpha- and beta-band *hypo*-synchrony during the preclinical stages was also observed in long-range synchrony (**Figure 3F and J**). Notably, the decreases in the long-range metrics were much greater than those in the local metrics, especially in the early stages during the phase of preclinical AD (stages 1–3).

## Discussion

We demonstrated that functional deficits of frequency-specific neural synchrony show progressive changes across AD stages. Both long-range and local neural synchrony in the alpha and beta bands, but not in the delta-theta band, was found to decrease in preclinical stages of AD, preceding neuro-degeneration and cognitive decline, with more robust findings for long-range neural synchrony. These findings highlight the frequency-specific manifestations of neural synchrony in AD and that synchrony reductions in the alpha and beta bands are sensitive indices reflecting functional deficits in the earliest stages of disease progression.

### Electrophysiological metrics of neural synchrony precede volume loss and cognitive decline

A key finding of the current study is that functional deficits as depicted by reduced neural synchrony precede structural volume loss and cognitive deficits. The EBMs on cross-sectional data clearly demonstrated that alpha- and beta-band synchrony within the inferior temporal and posterior parieto-occipital regions show significant deficits in the early disease stages–stages where volumetric and clinical deficits are still not significantly deviated from their baseline trajectory. This is consistent with the finding that functional changes occur earlier in the time course than structural changes in AD (*Jack et al., 2010*; *Sperling et al., 2011*).

Previous functional MRI studies have demonstrated disrupted connectivity especially between the hippocampus and several areas of the cortical default mode network (DMN) in subjects with amyloid deposition but without cognitive impairment (*Sperling et al., 2014*). This disruption in DMN has also been observed in clinically normal older individuals without prominent brain atrophy in MTL that preserves hippocampal activity (*Miller et al., 2008*; *Hedden et al., 2009*), indicating altered functional connectivity during the preclinical period of AD. In contrast to such fMRI data reflecting the cascade of neural, metabolic, hemodynamic events in AD, our findings from MEG, which captures the synaptic physiology as the collective oscillatory spectra, demonstrate direct observations of AD-related altered neuronal activity.

### Frequency-specific manifestations of neural synchrony deficits along the progression of the disease

We demonstrated that oscillatory deficits and their temporal association with neurodegeneration and cognitive decline are frequency-specific. In particular, it is the alpha and beta hyposynchrony that precedes PHG atrophy and MMSE decline, whereas the delta-theta hypersynchrony does not seem to show such a precedence. This is consistent with previous findings that alpha and beta hyposynchrony is more tightly associated with tau accumulation, which is closely allied to neurodegeneration and cognitive decline (*Pusil et al., 2019*; *Ranasinghe et al., 2020*; *Ranasinghe et al., 2021*). Neural

hyposynchrony in the alpha and beta bands may, therefore, represent harbingers of altered synaptic physiology associated with tau accumulation in AD. In fact, in human postmortem studies, the strongest correlate of cognitive deficits in AD patients is loss of synapse (*DeKosky and Scheff, 1990*; *Terry et al., 1991*). A study using transgenic AD mice has also shown that synaptotoxicity is an early phenomenon in AD pathophysiology (*Zhou et al., 2017*). In the context of fluid biomarkers to detect plasma amyloid, alpha and beta hyposynchrony can detect and quantify tau-associated neurodegenerative mechanisms, and hence may provide crucial information for early therapeutic interventions.

Previous studies have also shown that delta-theta oscillatory activity increases in AD and is strongly associated with amyloid accumulation (*Ranasinghe et al., 2020*; *Ranasinghe et al., 2022b*). In particular, increased delta-theta activity is a robust signal in individuals who are amyloid positive and cognitively unimpaired as well as those who harbor APOE-$\epsilon$4 allele and an increased risk of AD (*Cuesta et al., 2015*; *Nakamura et al., 2018*). These previous findings indicate that delta-theta hypersynchrony is an early change in the AD spectrum and may even precede neurodegeneration and cognitive deficits. However, in the current results, the trajectory of the delta-theta hypersynchrony was identical to or lagged that of the PHG volume loss. This apparent controversy may be due to the possibility that oscillatory changes in the delta-theta band are more closely related to amyloid accumulations in AD, which become saturated early in the disease course and have a poor association with neurodegeneration and cognitive trajectories. It would be worth exploring how the trajectory of early saturated variables may be captured by EBM approaches.

## Distinction between long-range and local synchrony deficits in disease progression

The decrease in alpha and beta-band long-range metrics in the preclinical stages was much greater than that in the local metrics. This is consistent with the fact that AD-related abnormal brain activities are observed as disruptions of functional networks. Long-range cross-regional metrics, such as AECs, directly capture network disruptions involving all brain regions, while local metrics capture features of individual regions. From the definition, local synchrony describes collective neuronal oscillations in each local region, and thus the change along AD progression may depend mainly on long-term, slowly changing regional neuronal loss. On the other hand, long-range synchrony describes temporal coherence amongst regional collective neuronal oscillations and is vulnerable to altered neuronal oscillations. Therefore, long-range metrics are more sensitive to abnormal rhythms, collecting local abnormalities.

Preclinical neurophysiological markers that indicate the pathophysiology of AD are clinically important but have not been established. Aβ accumulation in preclinical stages is just a necessary condition for AD, and additional preclinical markers are required to fully predict the progression of AD. From this point of view, the present study indicates that alpha- and beta-band MEG metrics, especially long-range synchrony metrics (AEC), which were found to be sensitive to preclinical stages, could be promising candidates as additional markers.

## Limitations

A limitation of the current study is that there were differences in age between controls and AD patients. Although we adjusted the age of each metric using GLMs, age trajectories in neurophysiological measures have been reported to be nonlinear even in healthy aging (*Sahoo et al., 2020*). Age-related changes in brain atrophy have also been reported to follow a nonlinear time course depending on the brain areas (*Coupé et al., 2019*). These studies indicate that it may be better to employ a non-linear method beyond GLM to perfectly correct aging effects.

Another limitation is that we have not performed independent validations of the predicted trajectories and also have not examined the heterogeneity in AD progression, although we clarified for the first time the time courses of MEG neurophysiological metrics in AD progression. In fact, AD is a heterogeneous multifactorial disorder with various pathobiological subtypes (*Jellinger, 2022*). In this context, an EBM called *Subtype and Stage Inference* (SuStaIn) capable of capturing spatio-temporal heterogeneity of diseases (*Young et al., 2018*) has been proposed to subtyping neurodegenerative diseases including typical AD and has been applied to find different spatio-temporal trajectories of longitudinal tau-PET data in AD (*Vogel et al., 2021*). Since oscillatory rhythms are thought to depend on AD subtypes (*Ranasinghe et al., 2017*; *Ranasinghe et al., 2022a*), an extended trajectory analysis

considering spatial and temporal variations of the MEG/EEG metrics is warranted in the future, and such analyses would provide distinct neurophysiological trajectories depending on AD subtypes. As a validation of the predicted trajectories, it would be necessary to investigate whether the predicted EBM stages are reliable and predictive of conversions (e.g. from control to MCI) while taking the AD subtypes into account.

## Acknowledgements

The authors thank all study participants for their support for our research.

## Additional information

### Competing interests

Kiwamu Kudo: KK is a full-time employee of Ricoh Company, Ltd. Hirofumi Morise: HM is a full-time employee of Ricoh Company, Ltd. Kensuke Sekihara: KS is an employee of Signal Analysis Inc. Srikantan S Nagarajan: SSN is a scientific consultant to MEGIN Inc and a Medical Strategy Adviser to Hippoclinic Inc. He served on the scientific advisory board for Rune Labs Inc from 2019-2022. He was the recipient of an industry contract from Ricoh MEG USA Inc. The other authors declare that no competing interests exist.

### Funding

| Funder | Grant reference number | Author |
| --- | --- | --- |
| National Institutes of Health | R01AG062196 | Srikantan S Nagarajan |
| National Institutes of Health | R01NS100440 | Srikantan S Nagarajan |
| National Institutes of Health | R01DC017091 | Srikantan S Nagarajan |
| National Institutes of Health | P50DC019900 | Srikantan S Nagarajan |
| National Institutes of Health | P30AG062422 | Bruce L Miller |
| National Institutes of Health | K23AG038357 | Keith Vossel |
| National Institutes of Health | K08AG058749 | Kamalini G Ranasinghe |
| National Institutes of Health | R21AG077498 | Kamalini G Ranasinghe |
| University of California | UCOP-MRP-17-454755 | Srikantan S Nagarajan |
| John Douglas French Alzheimer's Foundation | | Keith Vossel |
| S. D. Bechtel, Jr. Foundation and Stephen Bechtel Fund | | Keith Vossel |
| Alzheimer's Association | PCTRB-13-288476 made possible by Part the CloudTM | Keith Vossel |
| Alzheimer's Association | AARG-21-849773 | Kamalini G Ranasinghe |
| Larry L. Hillblom Foundation | 2015-A-034-FEL | Kamalini G Ranasinghe |
| Larry L. Hillblom Foundation | 2019-A-013-SUP | Kamalini G Ranasinghe |

| Funder | Grant reference number | Author |
| --- | --- | --- |
| Ricoh Company, Ltd. | Research contract | Heidi E Kirsch |

The funders had no role in study design, data collection and interpretation, or the decision to submit the work for publication.

## Author contributions

Kiwamu Kudo, Conceptualization, Software, Formal analysis, Investigation, Visualization, Methodology, Writing – original draft, Writing – review and editing; Kamalini G Ranasinghe, Conceptualization, Resources, Data curation, Formal analysis, Funding acquisition, Investigation, Visualization, Writing – original draft, Project administration, Writing – review and editing; Hirofumi Morise, Methodology, Writing – review and editing; Faatimah Syed, Data curation, Writing – review and editing; Kensuke Sekihara, Software, Methodology, Writing – review and editing; Katherine P Rankin, Joel H Kramer, Conceptualization, Resources, Supervision, Writing – review and editing; Bruce L Miller, Conceptualization, Resources, Supervision, Funding acquisition, Writing – review and editing; Gil D Rabinovici, Keith Vossel, Conceptualization, Resources, Data curation, Supervision, Funding acquisition, Investigation, Writing – review and editing; Heidi E Kirsch, Conceptualization, Supervision, Funding acquisition, Investigation, Writing – original draft, Writing – review and editing; Srikantan S Nagarajan, Conceptualization, Formal analysis, Supervision, Funding acquisition, Investigation, Visualization, Methodology, Writing – original draft, Project administration, Writing – review and editing

## Author ORCIDs

Kiwamu Kudo ⬡ https://orcid.org/0000-0002-5732-7229
Kamalini G Ranasinghe ⬡ https://orcid.org/0000-0002-4217-8785
Srikantan S Nagarajan ⬡ https://orcid.org/0000-0001-7209-3857

## Ethics

Human subjects: Informed consent was obtained from all participants and the study was approved by the Institutional Review Board (IRB) at UCSF (UCSF-IRB 10-02245).

Reviewer #1 (Public review): https://doi.org/10.7554/eLife.91044.3.sa1
Reviewer #2 (Public review): https://doi.org/10.7554/eLife.91044.3.sa2
Author response https://doi.org/10.7554/eLife.91044.3.sa3

# Additional files

## Supplementary files

• Supplementary file 1. Demographics and Neuropsychological assessments.

• Supplementary file 2. The 94 cortical/subcortical anatomical regions included in the Automated Anatomical Labeling 3 (AAL3) atlas.

• Supplementary file 3. Direct evaluation of the data likelihoods for all possible -score event sequences in the Atrophy-Cognition EBM (AC-EBM).

• Supplementary file 4. Top 10 regions with significant group differences in gray matter (GM) volumes comparison between Alzheimer's disease (AD) patients and controls.

• Supplementary file 5. Top regions with significant group differences in long-range synchrony between patients with Alzheimer's disease (AD) and controls.

• Supplementary file 6. Top 10 regions with significant group differences in local synchrony between patients with Alzheimer's disease (AD) and controls.

• Supplementary file 7. Top regions with significant weighted mean differences (false discovery rate, FDR corrected) in long-range synchrony between stages 4 and 1 (*Figure 2E* in the main text).

• Supplementary file 8. Top regions with significant weighted mean differences (false discovery rate, FDR corrected) in local synchrony between stages 4 and 1 (Figure 2F in the main text).

• Supplementary file 9. Pairs of stages with significant weighted-mean differences.

• Supplementary file 10. Top regions with significant weighted-mean differences (false discovery rate, FDR corrected) in regional variations of long-range synchrony during preclinical stages (stages

5 vs 1).

• Supplementary file 11. Pairs of stages with significant weighted-mean differences .

• Supplementary file 12. Top regions with significant weighted-mean differences (false discovery rate, FDR corrected) in regional variations of local synchrony during the preclinical stages (stages 6 vs 1) [Figure 4D, H, L in the main text].

• MDAR checklist

## Data availability

The processed datasets including PHG volumes, MMSE scores, scalar and regional MEG metrics, and a set of MATLAB scripts for reproducing all results and figures in the manuscript are available at OSF (https://doi.org/10.17605/OSF.IO/PD4H9). Any interested researcher can access the source data for all original figures in the manuscript by running the MATLAB scripts freely under the conditions of a CC0 license. De-identified MEG and MRI data are also shared on the OSF website. Please forward any correspondence and material requests to kamalini.ranasinghe@ucsf.edu or contact srikantan.nagarajan@ucsf.edu .

The following dataset was generated:

| Author(s) | Year | Dataset title | Dataset URL | Database and Identifier |
| --- | --- | --- | --- | --- |
| Kudo K, Ranasinghe K, Nagarajan S | 2024 | Neurophysiological trajectories in AD progression | https://doi.org/10.17605/OSF.IO/PD4H9 | Open Science Framework, 10.17605/OSF.IO/PD4H9 |

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
