## [Editor Report · eLife assessment]

This work presents **important** findings for the field of Alzheimer's disease, especially for the electrophysiology subfield, by investigating the temporal evolution of different disease stages typically reported using M/EEG markers of resting-state brain activity. The evidence supporting the conclusions is **convincing** and the methodology as well as the descriptions of the processes are of high quality, although a separation of individuals who are biomarker positive versus negative would have strengthened the results and conclusions of the study.

---

## [Referee Report · Reviewer #1 (Public review)]

Summary:

The authors aimed to infer the trajectories of long range and local neuronal synchrony across the Alzheimer's disease continuum, relative to neurodegeneration and cognitive decline. The trajectories are inferred using event-based models, which infer a set of data-driven disease stages from a given dataset. The authors develop an adapted event-based modelling approach, in which they characterise each stage as a particular biomarker increasing by a particular z-score deviation from controls. Fitting infers the optimal set of z-scores to use for each biomarker and the order in which each biomarker reaches each z-score. The authors apply this approach to data from 148 individuals (70 cognitively unimpaired older adults and 78 individual with mild cognitive impairment or Alzheimer's disease), identifying trajectories in which long-range (amplitude-envolope correlation) and local (regional spectral power) neuronal synchrony in the alpha and beta bands becomes abnormal prior to neurodegeneration (measured as the volume of the parahippocampal gyrus) and cognitive decline (measured using the mini-mental state examination).

Strengths:

- The main strength is that the authors assess two models. In the first they derive a staging system based only on the volume of the parahippocampal gyrus and mini-mental state examination score. They then investigate how neuronal synchrony metrics change compared to this staging system. In the second they derive a staging system that also includes an average (combined long-range and local) neuronal synchrony metric and investigate how long-range and local synchrony metrics change relative to this staging system. This is a strength as the first model provides confidence that there is not overfitting to the neuronal synchrony data, and the second provides more detailed insights into the dynamics of the early neuronal synchrony changes.

- Another strength is that the authors automatically infer the optimal z-scores to choose, rather than having to pre-select them manually, as in previous approaches.

Weaknesses:

- The authors do not have a dataset for external validation.

---

## [Referee Report · Reviewer #2 (Public review)]

Summary: This work presented by Kudo and colleagues is of great importance to strengthen our understanding of electrophysiological changes in the course of AD. Although the main conclusions regarding functional connectivity and spectral power change through the course of the disease are not new and have been largely studied and theorised on, this article offers an innovative approach that certainly consolidates previous knowledge on the topic. Not only that, this article also broadens our knowledge presenting useful and important details on the specificity of frequency and cortical distribution of these early alterations. The main take-home message of this work is the early disruption of electrophysiological signatures that precedes detectable alterations in other more commonly used pathology markers (i.e. gray matter atrophy and cognitive impairment). More specifically, these signatures include long-range connectivity in the alpha and beta bands, and local synchrony (spectral power) in the same frequency bands.

Strengths: The present work has some major strengths that make it paramount for the advance of our understanding of AD electrophysiology. It is a very well written manuscript that, despite the complexity of the analyses employed, runs the reader through the different steps of the analysis in a pedagogic and clever way, making the points raised by the results easy to grasp. The methodology itself is carefully chosen and appropriate to the nature of the question posed by the researchers, as event-based models are well-suited for cross-sectional data.

The quality of the figures is outstanding; not only are they aesthetic but, more importantly, the figures convey information exceptionally well and facilitate comprehension of the main results.

The conclusions of the paper are, in general, well described and discussed, and consider the state-of-the-art works of AD electrophysiology. Furthermore, even though the conclusions themselves are not groundbreaking at all (synaptic damage preceding structural and cognitive impairment is one of the epitomes of the pathological cascading model proposed by Jack in 2010), this article is innovative and groundbreaking in the way they address with clever analyses in a relatively large sample for neuroimaging standards.

Weaknesses: The authors increased the clarity of sample description after revisions (particularly control group characterization). However, even though it is true that a certain percentage of AB positivity is to be expected amongst cognitively healthy individuals, that doesn´t discard they are not expressing preclinical AD to some extent. I still feel that including only biomarker negative participants in the control group would increase the quality of the work. However, the sample is relatively well characterized as a whole and the results are interesting and in line with previous literature, thus limiting the apparent impact of these possible confounds.

---

## [Author Response]

The following is the authors’ response to the original reviews.

**eLife assessment**
This work presents important findings for the field of Alzheimer's disease, especially for the electrophysiology subfield, by investigating the temporal evolution of different disease stages typically reported using M/EEG markers of resting-state brain activity. The evidence supporting the conclusions is solid and the methodology as well as the descriptions of the processes are of high quality, although a separation of individuals who are biomarker positive versus negative would have strengthened the interpretability of the results and the conclusions of the study.

Response: Thank you for the positive assessment of the paper.

**Public Reviews:**

**Reviewer #1 (Public Review):**
Summary:The authors aimed to infer the trajectories of long range and local neuronal synchrony across the Alzheimer's disease continuum, relative to neurodegeneration and cognitive decline. The trajectories are inferred using event-based models, which infer a set of data-driven disease stages from a given dataset. The authors develop an adapted event-based modelling approach, in which they characterise each stage as a particular biomarker increasing by a particular z-score deviation from controls. Fitting infers the optimal set of z-scores to use for each biomarker and the order in which each biomarker reaches each z-score. The authors apply this approach to data from 148 individuals (70 cognitively unimpaired older adults and 78 individual with mild cognitive impairment or Alzheimer's disease), identifying trajectories in which long-range (amplitude-envolope correlation) and local (regional spectral power) neuronal synchrony in the alpha and beta bands becomes abnormal prior to neurodegeneration (measured as the volume of the parahippocampal gyrus) and cognitive decline (measured using the mini-mental state examination).Strengths:The main strength is that the authors assess two models. In the first they derive a staging system based only on the volume of the parahippocampal gyrus and mini-mental state examination score. They then investigate how neuronal synchrony metrics change compared to this staging system. In the second they derive a staging system that also includes an average (combined long-range and local) neuronal synchrony metric and investigate how long-range and local synchrony metrics change relative to this staging system. This is a strength as the first model provides confidence that there is not overfitting to the neuronal synchrony data, and the second provides more detailed insights into the dynamics of the early neuronal synchrony changes.Another strength is that the authors automatically infer the optimal z-scores to choose, rather than having to pre-select them manually, as in previous approaches.

Response: Thank you for the positive comments and a succinct summary of the paper and its strengths.

Weaknesses:The dataset is small and no external validation is performed.

Response: We agree that future validation studies of the predictions are necessary. We now include the related sentences in the last paragraph of the limitations section in the revised manuscript.

A high proportion of the data is from controls (nearly 50%) with no biomarker evidence of Alzheimer's disease, and so the changes may be driven by aging or other non-Alzheimer's effects.

Response: We would like to clarify that the z-scores of the metrics used in the EBMs were computed using age-adjusted values. All our controls were recruited from an ongoing longitudinal study of healthy aging. Amongst the 70 controls, 39 have confirmed A-beta negative PET scans and 8 were confirmed A-beta positive PET scans, and in the rest of the 23 we do not have any biomarker data available. However, in all the controls, we have conducted comprehensive neuropsychological assessment (see Appendix 1—table 1 in the revised supplementary file) and based on this data we can be quite confident about their lack of clinical deficits, and we have a very high degree of confidence that none of the controls have any neurodegeneration (AD-related or otherwise). Consistent with this assessment, in our EBM analyses, most of the control participants were indeed categorized to the preclinical stages.

Inferring the optimal z-scores is a strength, however as different sets of z-scores are allowed per biomarker, there is a concern that the changes reflected are mainly driven by the choice of z-score, rather than the markers themselves (e.g. if lower z-scores are selected for one marker than another, then changes in that marker will appear to be detected earlier, even if both markers change at the same time).

Response: Indeed, the biomarker sequence depends on the choice of the z-scores per biomarker. However, please note that our choice of z-scores is based on maximizing the sequence likelihood. Therefore, other values of the z-scores will have by construction a smaller likelihood of sequence occurrence compared to the results shown.

In equation 2 it is unclear why the gaussian is measured based on a sum over I. The more obvious choice would be to use a multivariate gaussian with no covariance, which would mean taking the product rather than the sum over I.

Response: We thank the reviewer for pointing this out and we now clarify this point. In this revision, we do not use the term ‘multivariate’. Indeed, the model likelihood assumes independence for each metric’s priors, and hence is the product of each metric’s univariate gaussian probability distribution. This can be seen in equations 1 and 2 of the revision manuscript (Section titled “Event-based sequencing modeling’). The assumption about independent priors is similar to the one used in the original event-based model (see equation (2) in A .L. Young et al., Nature Comm. 9.1 (2018): 4273).

In the original event-based model, k is a hidden variable. Presumably that is also the case here, however the notation k=stage(j) makes it seem like each subject is assigned a stage during the sequence optimisation.

Response: We would like to clarify that the posterior probability of each stage for every subject is estimated during the sequence optimization. To clarify the notation, we have now deleted the term “stage” and use “tj” to denote stages for each subject j. The sequence optimization was performed with the assumption of a uniform prior distribution p(tj=k) = 1/(N+1) for each stage k. Then, the posterior probability p(tj=k|Zj,S), i.e., the probability that subject j belongs to stage k, given the metrics and the sequence, was computed during the sequence optimization procedure.

Typically for event-based modeling, positional variance diagrams are created from the markov chain monte carlo samples of the event sequence, enabling visualisation of the uncertainty in the sequence, but these are not included in the study.

Response: In the revised supplementary file, we have now included positional uncertainty diagrams for the optimal set of z-score events that were created from 50,000 MCMC samples. Please see Appendix 1—ﬁgure 2 for the AC-EBM and Appendix 1—ﬁgure 9 for the SAC-EBMs.

Many of the figures in the manuscript (e.g. Figure 1E/G, Figure 2A/B, Figure 3A/B/E/F/I/J, Figure 4 A/B/E/F/I/J) are based on averages in both the x and the y axis. In the x dimension, individuals have a weighted contribution to the value on the y axis, depending on their stage probability. In the y dimension, the values are averages across those individuals, and the error bars represent the standard error rather than the standard deviation. Whilst the trajectories themselves are interesting, they may not be discriminative at the individual level and may be more heterogeneous than it appears.

Response: In the current study, the predictions of trajectories are intended at the cohort level. Individual level investigations will be the topic of future investigations.

The bootstrapped statistical analyses comparing metrics between the stages do not consider the variability in the sequence.

Response: Please see the response above. The positional uncertainty diagrams are included in the revised supplementary file.

**Reviewer #2 (Public Review):**
Summary:This work presented by Kudo and colleagues is of great importance to strengthen our understanding of electrophysiological changes in the course of AD. Although the main conclusions regarding functional connectivity and spectral power change through the course of the disease are not new and have been largely studied and theorised on, this article offers an innovative approach that certainly consolidates previous knowledge on the topic. Not only that, this article also broadens our knowledge presenting useful and important details on the specificity of frequency and cortical distribution of these early alterations. The main take-home message of this work is the early disruption of electrophysiological signatures that precedes detectable alterations in other more commonly used pathology markers (i.e. gray matter atrophy and cognitive impairment). More specifically, these signatures include long-range connectivity in the alpha and beta bands, and local synchrony (spectral power) in the same frequency bands.

Response: Thank you for the positive comments and for providing a nice succinct summary.

Strengths:The present work has some major strengths that make it paramount for the advance of our understanding of AD electrophysiology. It is a very well written manuscript that, despite the complexity of the analyses employed, runs the reader through the different steps of the analysis in a pedagogic and clever way, making the points raised by the results easy to grasp. The methodology itself is carefully chosen and appropriate to the nature of the question posed by the researchers, as event-based models are well-suited for cross-sectional data.The quality of the figures is outstanding; not only are they aesthetic but, more importantly, the figures convey information exceptionally well and facilitate comprehension of the main results.The conclusions of the paper are, in general, well described and discussed, and consider the state-of-the-art works of AD electrophysiology. Furthermore, even though the conclusions themselves are not groundbreaking at all (synaptic damage preceding structural and cognitive impairment is one of the epitomes of the pathological cascading model proposed by Jack in 2010), this article is innovative and groundbreaking in the way they address with clever analyses in a relatively large sample for neuroimaging standards.

Response: Thank you for the positive comments of the strengths of the paper.

Weaknesses:The main limitation of the work revolves around sample definition and inclusion criteria that are somewhat confusing obscuring some of the points of the analyses. Firstly it is not clear why the purely clinical approach is employed to diagnose the "probable Alzheimer´s Disease" for the 78 participants in the "AD group". In the same paragraph, it is stated that 67 out of the 78 participants show biomarker positivity, thus allowing a more biologically guided diagnosis that is preferred according to current NIA-AA criteria. This would avoid highly possible mixing of different subtypes of dementia etiologies. One might wonder, why would those 11 participants be included if we have strong indications that their symptoms are not due to AD? Furthermore, the real pathological status of the control group is somewhat questionable. The authors do not specify whether common AD biomarkers are available for this subgroup. In that case, it would have highly increased the clarity and interpretability of the results if this group was subdivided in a preclinical and completely healthy control group. This would be particularly interesting since a significant proportion of the control group is labeled as belonging to stages 2,3,4 (MCI) and even 5 (mild dementia). This raises the question of whether these participants are true healthy controls mislabeled by the EBM model, or actual cognitive controls with actual underlying AD pathology well identified by the model proposed.

Response: Please see responses above to a similar comment from R1. To clarify, all our controls were recruited from an ongoing longitudinal study of healthy aging. Amongst the 70 controls, 39 have confirmed A-beta negative PET scans and 8 were confirmed A-beta positive PET scans, and in the rest of the 23 we do not have any biomarker data available. The biomarker positivity rates in our control cohort are completely consistent with the prevalence of A-beta positivity in cognitively healthy individuals and are within a normal biological continuum for amyloid beta (Jansen WJ et al. 2015). In all the controls, we have conducted comprehensive neuropsychological assessment (see Appendix 1—table 1 in the revised supplementary file) and based on this data we can be quite confident about their lack of clinical deficits, and we have a high degree of confidence that none of the controls have any neurodegeneration (AD-related or otherwise). We include these details in the revision (see the revised ‘Participants’ section in the Materials and methods.).

Jansen WJ et al., 2015 JAMA; 667 313(19):1924-1938.

On this note, Figure 2 (C and D) and Figure 3 (C, G and K) show a cortical surface depicting the mean difference of each stage vs the control group, which again, is formed by subjects that can be included (and in fact, are included) in all those stages, obscuring the meaning and interpretability of these cortical distributions.

Response: We would like to clarify that these figures depict the regional maps of each metric for each stage of AD progression, not the contrast against a control group.

**Reviewer #1 (Recommendations For The Authors):**
If possible, perform independent validation of the results.

Response: This is something we indeed intend to examine in our future investigations.

Repeat the analysis in the subset of individuals that are amyloid positive.

Response: Amongst the 78 AD patients, 20 had autopsy confirmed AD neuropathology, an additional 41 patients had molecular pathology identified by Abeta-PET, and another additional 9 had fluid biomarker (CSF) confirmation of amyloid and tau levels consistent with AD diagnosis. Eight remaining patients had a diagnosis of AD with high certainty, based on clinical presentation, neurological assessment, and cortical atrophy on MRI. Given that there are only eight patients who had clinical diagnosis of AD (with no biomarkers), and the comprehensive clinical characterization of all the AD patients in our cohort (Appendix 1—table 1), we do not believe that any subgroup analysis is warranted.

When inferring the optimal z-scores, select the same set of z-scores per biomarker, or include diagrams of stage vs z-score that include all of the markers so that it is easy to see how one marker changes relative to the others (overlay Figure 1G on Figure 2A and 2B).

Response: How the neural synchrony metrics, PHG volume and MMSE scores change relative to each other is exactly what we show in Figures 3 B/F/J and 4 B/F/J. Since each EBM model optimizes the z-score thresholds, sequence likelihood and posterior probability of each stage for each subject, the EBM framework provides the most likely estimate for each metric at every stage. Therefore, the SAC-EBM model gives the most accurate description of the relative differences in these metrics over the AD progression stages. The reviewer’s suggestion to overlay Figure 1G (now figure 1F, based on optimized z-scores for PHG volume and MMSE scores) on Figures 2A and 2B will be inaccurate, as the neural synchrony measures plotted in figures 2A and 2B are not for optimized z-scores.

Change equation 2 to use a multivariate gaussian.

Response: We now clarify that we use a factorized multivariate form that reflects independent priors for each metric which are Gaussian.

Clarify whether k is a hidden variable and possibly change the notation.

Response: We now clarify that in our notation, k is a label for the stage [k=1,..,7 (when I=2) or k=1,...,10 (when I = 3)] and is indeed a hidden variable and not observed (but inferred from the EBM). Specifically, the posterior probability for each subject j belonging to stage k was estimated as part of the sequence optimization procedure.

Generate positional variance diagrams of the MCMC samples.

Response: We are doing the MCMC to obtain the most likely sequence. We have now included positional variance diagrams of the optimal set of z-score events in Appendix 1—ﬁgure 2 and Appendix 1—ﬁgure 9 in the revised supplementary file.

It would be interesting to study whether the stages are predictive of conversion or look at longitudinal data, if available.

Response: This is something we indeed intend to examine in our future investigations.

Also look at statistics across MCMC samples of the sequence.

Response: Thank you for this suggestion. In the Appendix 1—ﬁgure 10, we now include an example of the MCMC samples for an SAC-EBM including the alpha-band AEC. We then derived the positional variances for each metric that are now shown in Appendix 1—ﬁgure 2 and Appendix 1—ﬁgure 9.

**Reviewer #2 (Recommendations For The Authors):**
Some really minor changes are suggested on two specific points that somewhat confused me as a reader and got me stuck in the reading process to try to get the meaning of what I was seeing/reading:1. It is not specified (or at least I was unable to find it) what are you comparing exactly for the group comparison in the long-range synchrony metric (AEC) before creating your scalar metric. Are you comparing individual links (in which case you would have 93 link values for each ROI to compare)? Or are you comparing the strength for each ROI (thus, one value -the individual links sum- for each ROI)? I guess it should be the latter for what I see in the figures but it could be useful to specify it.

Response: The reviewer is correct. We compare the strength of each ROI, i.e., averaging over edges of the symmetric AEC matrix of functional connectivity. We now clarify this in the Amplitude-envelope correlation section and the caption of the revised Appendix 1—ﬁgure 6.

1. In Figure 1 (which, by the way, is exceptionally aesthetic, congratulations for that!) I got stuck for a relatively long time in a really small detail and I am not completely sure if I came to the right conclusion. It is regarding the X axis of the histograms in panels B and D. They are expressed as "PHG volume loss" and "MMSE decline". So I supposed those histograms were showing some kind of subtraction, (maybe from stage X to stage Y, or from group X to group Y). I was trying to understand the histogram and rereading methods to see if I overlooked any description of that graphic and then just realized they might be just the Z-score itself for each group (control and AD) with respect to the whole population. If that is the case I would suggest changing the X-label to "PHG z-score" and "MMSE z-score" avoiding the reference to "loss and "decline" as they are just reflecting the direct transformation to z-score.

Response: Thank you. We would like to clarify that the z-score for PHG volume and MMSE scores were sign-inverted so that higher values denote “PHG Volume loss” and “MMSE decline”, respectively. We now clarify this point in the revised text and legend for the revised figure 1.

Lastly, regarding the point I raised in the limitations section of the public review, I understand it might fall out of the scope of eLife reviewing process as it would require a more extensive change of the current manuscript, which is great as it is. But as a reader and researcher in the field, I would have recommended using biomarkers to divide the control group (if available) thus including in the models only those belonging to the AD continuum according to their biomarker status, and leaving those control without any biomarker positivity as the reference group for the figures I mention in that section (those showing differences for each stage in the cortical surface with respect to the control group).

Response: Please see a similar comment from R1. Amongst the 70 controls, 39 have confirmed A-beta negative PET scans and only 8 were confirmed A-beta positive PET scans, and in the rest of the 23 we do not have any biomarker data available. In all the controls, we have conducted comprehensive neuropsychological assessment (see Appendix 1—table 1 in the revised supplementary file) and based on this data we can be quite confident about their lack of clinical deficits, and we have a high degree of confidence that none of the controls have any neurodegeneration (AD-related or otherwise). Since only 8 participants were confirmed as amyloid positive in the control group and this sample size is small, we do not conduct this recommended re-analysis in this manuscript.